# Incorporating domain knowledge into neural-guided search via *in situ* priors and constraints

**Brenden K. Petersen**[*][†]                                          BP@LLNL.GOV
**Claudio P. Santiago**[*]                                          PRATA@LLNL.GOV
**Mikel Landajuela**                                LANDAJUELALA1@LLNL.GOV
*Lawrence Livermore National Laboratory*

## Abstract

Many AutoML problems involve optimizing discrete objects under a black-box reward. Neural-guided search provides a flexible means of searching these combinatorial spaces using an autoregressive recurrent neural network. A major benefit of this approach is that builds up objects *sequentially*—this provides an opportunity to incorporate domain knowledge into the search by directly modifying the logits emitted during sampling. In this work, we formalize a framework for incorporating such *in situ* priors and constraints into neural-guided search, and provide sufficient conditions for enforcing constraints. We integrate several priors and constraints from existing works into this framework, propose several new ones, and demonstrate their efficacy in informing the task of symbolic regression.

## 1. Introduction

> *"Any practical algorithm must avoid exploring all but a tiny fraction of the state space."*
> — Artificial Intelligence: A Modern Approach
> Russell and Norvig (2002)

Many problems in automated machine learning (AutoML) fall into the category of *symbolic optimization* (SO). We consider the following discrete optimization problem:

$$\underset{n \in \mathbb{N}, \tau_1, \ldots, \tau_n}{\arg \max} \ [R(\tau_1, \ldots, \tau_n)] \ \text{ with } \tau_i \in \mathcal{L} = \{\alpha, \beta, \ldots, \zeta\}$$

In this formulation, $\tau = [\tau_1, \ldots, \tau_n]$ is a discrete object represented by a variable-length sequence of discrete symbols or "tokens" $\tau_i$ selected from a library $\mathcal{L}$, and $R$ is a black-box (i.e. non-differentiable) reward function. A popular SO problem is neural architecture search (NAS), in which tokens represent architectural hyper-parameters, the sequence represents a specification of a neural network

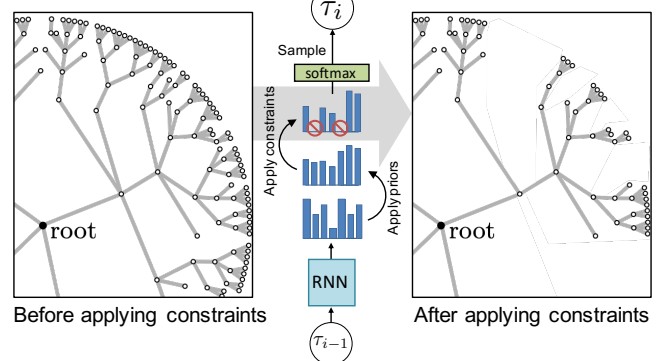

Before applying constraints | After applying constraints

Figure 1: Overview: Pruning the search tree in neural-guided search via *in situ* constraints.

---

∗. Equal contribution.
†. Corresponding author.

architecture, and the reward is the validation accuracy when instantiating the specified network and training it on some downstream task (Zoph and Le, 2016). Other examples include program synthesis (Abolafia et al., 2018), symbolic regression (Petersen et al., 2021), *de novo* molecular design (Popova et al., 2019), automated theorem proving (Bibel, 2013), and many traditional combinatorial optimization problems (e.g. traveling salesman problem) (Bello et al., 2016).

For many SO problems, the object is not traditionally represented as a sequence; however, these can be reduced to SO by establishing a sequential representation of the object. For example, small molecules can be represented by simplified molecular-input line-entry system (SMILES) strings (Weininger, 1988), and mathematical expressions can be represented by their pre-order traversals.

SO is challenging due to its combinatorial nature, as the size of the search space grows exponentially with the length of the sequence. Because of this, it is commonly solved using evolutionary approaches (e.g. genetic programming), which are easily scalable and broadly applicable (Banzhaf et al., 1998). More recently, *neural-guided search* has proven to be a successful alternative (Zoph and Le, 2016; Bello et al., 2016). In this framework, an autoregressive recurrent neural network (RNN) generates objects sequentially. Specifically, the RNN emits a vector of logits $\ell$ that defines a categorical distribution over tokens in the library $\mathcal{L}$, conditioned on the previously sampled tokens via the RNN state. A token is then sampled: $\tau_i \sim p(\tau_i|\tau_{1:(i-1)}; \theta) = \text{Categorical}(\text{Softmax}(\ell))$, where $\theta$ are the RNN parameters. The RNN is trained using reinforcement learning or other heuristic methods (Abolafia et al., 2018) aimed at increasing the likelihood of high-reward objects.

A key methodological aspect of autoregressive approaches is that they generate objects *sequentially*, i.e. one token at a time. In contrast, evolutionary approaches generate new objects by making edits (via mutations and crossovers) to existing objects. A benefit of sequential object generation is that it affords an opportunity to incorporate prior knowledge into the search phase during each step of the generative process. More specifically, one can incorporate knowledge *in situ* (i.e. during the autoregressive generative process) by directly modifying the emitted logits $\ell$ before sampling each token. In contrast, non-sequential approaches resort to *post hoc* methods such as rejection sampling, which is inefficient and can lead to intractable likelihoods.

As a simple example, consider expert knowledge that the current token should not be the same as the previous token, i.e. we want to impose the constraint $\tau_i \neq \tau_{i-1}$. This particular constraint can be incorporated by simply adding negative infinity to the logit corresponding to $\tau_{i-1}$ each step of autoregressive sampling.

While several individual works incorporate priors and/or constraints into their search, to our knowledge there is no existing work that formalizes this approach. Thus, we make the following contributions: (1) defining a simple yet flexible framework for imposing *in situ* priors and constraints for autoregressive neural-guided search, (2) formalizing sufficient conditions for imposing *in situ* constraints, (3) providing a centralized list of previously proposed priors and constraints, (4) proposing several new priors and constraints, and (5) empirically demonstrate their efficacy in the task of symbolic regression.

Our overarching goal is to provide a set of priors and constraints that is sufficiently broad to spark new ideas and encourage readers to design their own novel priors and constraints suited to their particular AutoML tasks.

## 2. Related Work

In many areas of AI, the idea of *pruning*, or eliminating possibilities from consideration without having to examine them, is critically important (Russell and Norvig, 2002). In adversarial search for games, where branching factors are high and exploring all possible moves is infeasible, alpha-beta pruning (Edwards and Hart, 1961) can be used to eliminate branches of the game tree that have no effect on the final evaluation and would be otherwise futilely evaluated by the standard minimax algorithm. Other approaches, like forward pruning (Greenblatt et al., 1967) or futility pruning (Heinz, 1998), reduce the space by eliminating moves that appear to be poor moves based on heuristic evaluation functions. The recent achievements in the game of Go (Silver et al., 2016) are also largely attributed to using policy and value networks (trained over millions of games) to guide Monte Carlo Tree Search (MCTS) rollouts, thereby avoiding exploring paths that would result in sub-optimal moves while focusing computational resources in exploring the most promising paths.

In the field of SO, Popova et al. (2019) use valency constraints to follow rules of organic chemistry to optimize molecular structures. Li et al. (2019) use a neural-guided approach to identify asymptotic constraints of leading polynomial powers and use those constraints to guide MCTS for the problem of symbolic regression. Kim et al. (2021b) employ several expression-based constraints to inform the search for symbolic control policies in reinforcement learning.

## 3. A Framework for *in situ* Priors and Constraints

In this work, we define a prior as a *logit adjustment vector*, denoted $\ell_\circ$, that is added to the emitted RNN logits $\ell$ during autoregressive sampling: $\tau_i \sim \text{Categorical}(\text{Softmax}(\ell + \ell_\circ))$. We define constraints as a special case of "hard" priors in which logit adjustments are either zero (no effect) or negative infinity (the token cannot be selected), denoted $\ell_\oslash$. Thus, priors *bias* the search but do not reduce the size of the search space, whereas constraints *prune* (i.e. eliminate possible sequences of) the search space. Typically, multiple priors and constraints can be composed simply by summing their logit adjustments, yielding the final distribution for the $i$th token: $\tau_i \sim \text{Categorical}(\text{Softmax}(\ell + \sum_j \ell_\circ^{(j)} + \sum_k \ell_\oslash^{(k)}))$, where $\ell_\circ^{(j)}$ is the $j$th prior and $\ell_\oslash^{(k)}$ is the $k$th constraint.

Tables 1 and 2 provide succinct summaries of several broadly applicable classes of constraints and priors, respectively. In the sections below, we provide details for three novel constraints, two additional constraints which appear in existing works but we generalize in this work, and two novel priors. Detailed descriptions of other existing constraints and priors are provided in the Appendix.

### 3.1 Example classes of *in situ* constraints

**Lexicographical constraint.** Many SO problem exhibit large *semantic equivalence classes*, or sets of objects whose semantics are identical, e.g. $x + y$ and $y + x$ in mathematics, $CH_4$ and $H_4C$ in chemistry, or $A \wedge B$ and $B \wedge A$ in logical reasoning. To reduce the number of semantically equivalent sequences generated for tree-based objects, we can assign a predefined, arbitrary order for the operands of commutative operators (e.g. $+$ or $\times$). To do so, we posit a lexicographical ordering of the tokens, defined by an injective function $l(\tau_i)$, where

$l : \mathcal{L} \to \mathbb{N}$. Given a commutative operator token of arity $n$, the lexicographical constraint is enforced by requiring the lexicographical value of a child token to be no less than that of the previous child token, i.e. children are lexicographically sorted. Given $n$ different child tokens, there are $n!$ possible ways to place them as children of an $n$-ary commutative operator. This constraint reduces the number of possible orderings to one. Further, since this reduction propagates through the subtrees, it reduces the search space exponentially with the number of commutative tokens. Note that this constraint reduces the size of the search space without reducing the number of *semantically unique* objects in the search space. *Example.* Under the lexicographical constraint applied to the commutative operator $+$, if $l(\cos) > l(\sin)$, then the expression $\cos(\square) + \sin(\square)$ (in this order) is constrained but $\sin(\square) + \cos(\square)$ is allowed.

**Subtree length constraint.** Similar to the lexicographical constraint, the subtree length constraint reduces the number of semantically repeated sequences generated. Here it does so by limiting the length of a subtree by the length of the preceding sibling subtree. When sampling a tree-based object by its pre-order traversal, a given subtree completes before the next sibling subtree begins. Thus, we can implement constrain each subtree to have a maximum length given by preceding sibling subtree length. *Example.* Under the subtree length constraint applied to the commutative binary operator $+$, consider the partial sequence $[+, \sin, x]$ (corresponding to the expression $\sin(x) + \square$). The first subtree ($\sin(x)$) has length 2, so the next subtree ($\square$) length is constrained to be at most 2.

**Type and unit constraints.** In NAS, different positions along the sequence have different types, (e.g. activation function, number of neurons), each of which has its own set of allowable tokens. A type constraint ensures that all positions satisfy the specified types. More generally, as in the case of mathematical expressions, tokens may have specific input/output types or units. For example, the $\times$ token has no requirements on input units, but the output units must be equal to the product of the input units. *Example.* Given the partial sequence $[\times, x]$ (corresponding to the expression $x \cdot \square$), where $x$ has units kg and the final output has units $\text{kg}^2$, tokens whose output types cannot be kg are constrained (e.g. trigonometric functions, input variables with units other than kg).

**Relational constraint.** A broad class of constraints for tree-based objects involves preventing one arbitrary set of tokens (called "targets") from having a particular structural relationship with another arbitrary set of tokens (called "effectors"). Common relationships include descendants, children, and siblings. This type of constraint is useful in expression search spaces for preventing nested trigonometric functions, inverse unary operators cancelling out (e.g. $\log(\exp(x))$), or redundantly adding together two constants. A few instances of relational constraints are used in Petersen et al. (2021); here, we generalize them into a single constraint class. *Example.* Under the relational constraint "[trigonometric functions] cannot be the [descendant] of [trigonometric functions]," given the partial sequence $[\sin, +, x, \times, x]$ (corresponding to the expression $\sin(x + x \cdot \square)$), all trigonometric tokens are constrained because they are a descendant of sin.

**Blacklist constraint.** For all object types, one can constrain a specified set of "blacklisted" sequences (or partial sequences) from being sampled. This may be useful to prune the search space of previously known sub-optimal solutions. *Example.* Liang et al. (2018) implement a special case of this constraint called "systematic exploration" in which the set

Table 1: Descriptions of various constraints. Objects (Obj.) refers to the types of objects to which the constraint applies (i.e. any, tree-based, molecule). References (Ref.) are [1] Petersen et al. (2021), [2] Kim et al. (2021b), [3] Liang et al. (2018), and [4] Popova et al. (2019). $\star$: The reference introduces a special case, which we generalize in this work.

| Constraint | Description | Obj. | Ref. |
|---|---|---|---|
| Length | *"Sequences must fall between* [min] *and* [max] *length."* | Any | [1] |
| Relational | *"*[Targets] *cannot be the* [relationship] *of* [effectors].*"* | Tree | [1]$^\star$ |
| Repeat | *"*[Target tokens] *must appear between* [min] *and* [max] *times."* | Any | [1, 2] |
| Blacklist | *"Sequences already in* [buffer] *are constrained."* | Any | [3]$^\star$ |
| Valency | *"Atoms must adhere to valency rules."* | Mol. | [4] |
| Lexicographical | *"Children of* [target token] *must be lexicographically sorted."* | Tree | Here |
| Subtree length | *"Subtrees of* [target token] *must be sorted by length."* | Tree | Here |
| Type & Unit | *"All tokens must follow specified types and/or units."* | Any | Here |

of blacklisted sequences grows over time, and is defined by the history of sequences sampled so far. Under this constraint, the same sequence will never be sampled more than once.

### 3.2 Sufficient conditions for imposing *in situ* constraints

Here we describe sufficient conditions for the ability to enforce a particular *in situ* constraint. Let $\mathcal{C}_i$ be the desired set of constrained/disallowed tokens before sampling $\tau_i$. Autoregressive sampling can enforce any constraint that can be expressed as a function $f(\cdot)$ of the partial sequence $\tau_{1:i-1}$, the library of tokens $\mathcal{L}$, and any constraint parameters $\Omega$ (e.g. [min] and [max] for the length constraint), provided that $\mathcal{C}_i$ does not include all tokens in $\mathcal{L}$. That is, if there exists an $f(\cdot)$ such that $\mathcal{C}_i = f(\tau_{1:i-1}, \mathcal{L}, \Omega)$ and $\mathcal{C}_i \subsetneq \mathcal{L}$, then autoregressive sampling can enforce the constraint induced by $f(\cdot)$. Further, $k$ independent constraints $\mathcal{C}_i^{(1)}, \ldots, \mathcal{C}_i^{(k)}$ can be combined if and only if $\mathcal{C}_i^{(1)} \cup \cdots \cup \mathcal{C}_i^{(k)} \subsetneq \mathcal{L}$. Notably, these conditions preclude constraints based on the semantics or knowledge of the complete sequence $\tau$. For example, one cannot constrain the search to all expressions with a particular range, or to molecules with a specific melting point. We describe several examples under this formalism:

*Lexicographical*: $C_i = \{v : l(v) < l(\tau_{\text{eft}}), v \in \mathcal{L}\}$, where $\tau_{\text{eft}}$ is the left sibling of $\tau_i$. The constraint parameters are the lexicographical values: $\Omega = \{l(v) : v \in \mathcal{L}\}$.

*Subtree length*: Set $N \leftarrow i + L(\tau_{1:i-1})$ and apply the *length constraint* with [min] = 1 and [max] = N, where $L(\tau_{1:i-1})$ is the size of the subtree rooted at the left sibling of $\tau_i$.

*Blacklist*: $\mathcal{C}_i = \{v : (\tau_{1:i-1} \| v) \in \mathcal{T}, v \in \mathcal{L}\}$, where $\mathcal{T}$ is the set of blacklisted sequences and $\|$ represents concatenation. In this case, $\Omega = \mathcal{T}$.

### 3.3 Example classes of *in situ* priors

**Token-specific priors.** Token-specific priors allow users to differentially control the relative prior probability of particular tokens. Specifically, the user selects a vector $\lambda$ of length $|\mathcal{L}|$ that specifies the desired relative prior probability of each token. The token-specific prior adds logits $\ell_\circ$ such that the resulting probability vector is multiplied element-wise by $\lambda$ and renormalized to sum to unity. To compute this prior, we seek $\ell_\circ$ such that $\text{Softmax}(\ell + \ell_\circ) = (\lambda \odot \text{Softmax}(\ell))/(\lambda \cdot \text{Softmax}(\ell))$, where $\odot$ represents element-wise mul-

Table 2: Descriptions of various priors. References (Ref.) are [5] Landajuela et al. (2021) and [6] Kim et al. (2021a).

| Prior | Description | Obj. | Ref. |
|---|---|---|---|
| Soft length | *"Sequences are discouraged to have length far from* [length]*."* | Tree | [5] |
| Uniform arity | *"The prior probability over* arities *is uniform."* | Tree | [5] |
| Language model | *"Probabilities are informed by language model outputs."* | Any | [6] |
| Token-specific | *"Tokens have a relative fold-increase prior probability of* [$\lambda$]*."* | Any | Here |
| Positional | *"Tokens at position* [i] *follow a token-specific prior."* | Seq. | Here |

tiplication and $\cdot$ represents the dot product. The solution is simply $\ell_\circ = \log \lambda$ (defined up to an additive constant); interestingly, this does not depend on $\ell$.

**Positional priors.** For sequence-based objects, it may be desirable to apply different token-specific priors at different positions along the sequence. In this case, separate token-specific priors can be applied at particular positions along the sequence. For example, in NAS, given a known, high-performing reference architecture, one can guide the search toward similar architectures by applying positional priors that bias each position toward the corresponding value in the reference architecture.

## 4. Experiments and Discussion

For empirical analysis, we consider *symbolic regression*, the task of discovering tractable mathematical expressions to fit a dataset $(X, y)$, where $X \in \mathbb{R}^n$ and $y \in \mathbb{R}$. In this SO problem, tokens represent mathematical operators (e.g. $\sin, \times$). A sequence $\tau$ is instantiated as a function $\tau = f(X)$, where $f : \mathbb{R}^n \to \mathbb{R}$, which is used to predict values $\hat{y} = f(X)$. Reward is based on mean-square error between $\hat{y}$ and $y$.

Table 3: Average recovery rate and steps to solve across the 12 Nguyen benchmarks ($n = 20$).

| | DSR | | Random search | |
|---|---|---|---|---|
| **Experiment** | **Recovery** | **Steps** | **Recovery** | **Steps** |
| No $\ell_\circ, \ell_\oslash$ | 57.5% | 1069.0 | 14.2% | 1785.9 |
| Lexicographical | 71.7% | 801.9 | 22.5% | 1643.8 |
| Subtree length | 66.3% | 871.4 | 20.8% | 1693.6 |
| Trigonometric | 83.3% | 519.6 | 21.3% | 1700.7 |
| Inverse | 57.9% | 1054.6 | 13.3% | 1792.3 |
| Soft length | 75.4% | 701.2 | 46.7% | 1298.6 |
| Max length | 59.6% | 1148.3 | 17.5% | 1759.2 |
| All $\ell_\circ, \ell_\oslash$ (L) | 84.2% | 552.0 | 52.9% | 1107.0 |
| All $\ell_\circ, \ell_\oslash$ (S) | 83.8% | 611.1 | 55.4% | 1111.1 |

Table 3 shows performance on the Nguyen symbolic regression benchmarks (Uy et al., 2011), using either deep symbolic regression (DSR) (Petersen et al., 2021) or random search, i.e. DSR with learning rate set to 0. We observe that adding individual priors or constraints (rows $2 - 7$) generally improves performance over no priors or constraints (row 1), for both DSR and random search. Combining all priors and constraints (rows 8 and 9; note that the lexicographical constraint (L) and subtree length constraint (S) are mutually incompatible) greatly improves performance. Lastly, it is interesting to note that random search with all priors and constraints (55.4%) achieves nearly the same performance as DSR with no priors or constraints (57.4%), demonstrating the ability of priors and constraints to effectively bias and prune the search.

## 5. Conclusion

We consider a generic framework for incorporating *in situ* priors and constraints into neural-guided search, which we believe is well-suited for integrating expert knowledge into symbolic optimization tasks. By contextualizing many existing priors and constraints within this framework and proposing several new ones, we hope to encourage researchers to design their own priors and constraints suited to their particular AutoML tasks.

## Acknowledgments

This work was performed under the auspices of the U.S. Department of Energy by Lawrence Livermore National Laboratory under contract DE-AC52-07NA27344. Lawrence Livermore National Security, LLC. LLNL-CONF-822798.

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

## Appendix A.

### A.1 Descriptions of existing constraints

To describe existing constraints, we define $\delta$ as the delta function, $d_i$ as the number of uns-elected or "dangling" nodes at position $i$ (for tree-based objects), $\mathbb{1}_{\text{condition}}$ as the indicator function (which returns 1 if condition is true and 0 otherwise), and $\mathcal{A}(n)$ as the set of tokens in $\mathcal{L}$ with arity $n$ (for tree-based objects).

**Length constraint.** Many SO problems allow variable-length objects, e.g. *de novo* molecular design, program synthesis, symbolic regression, and some formulations of NAS. One can constrain the length to prevent objects that are either too complicated or too simple. An interesting special case of this constraint is when the minimum and maximum length are equal. For tree- or graph-based objects, this constrains the search to different structures of the same length. This can be used for the task of searching over the space of molecular isomers, e.g. straight-chain alkanes. Given constraint parameters $\Omega = \{[\min], [\max]\}$, the length constraint is given by:

$$\mathcal{C}_i = \{u : u \in \mathcal{A}(0)\}\mathbb{1}_{d_i=1} \cup \{u : u \in \cup_{x>0}\mathcal{A}(x)\}\mathbb{1}_{d_i+i=[\max]}.$$

*Example 1.* Given the partial sequence $[\sin, +, x]$ (corresponding to the expression $\sin(x + \square))$ with a maximum length of 5, all tokens with arity greater than 1 are constrained because the sequence would no longer be able to complete before exceeding the maximum length.

*Example 2.* Given the partial sequence $[+, x]$ (corresponding to the expression $x + \square$) with a minimum length of 4, terminal tokens are constrained because they would complete the sequence prematurely.

**Repeat constraint.** Given a set $\mathcal{T}$ of target tokens and a $[\min]$ and $[\max]$ value, the repeat constraint aims to limit the number of times each target token appears to fall between $[\min]$ and $[\max]$, inclusive. Set $\mathcal{N}_v \leftarrow \sum_{k=1}^{i-1} \delta_{\tau_k,v}, \forall v \in \mathcal{T}$. Then, for each target token $v \in \mathcal{T}$, set

$$\mathcal{C}_i^{(v)} = \{u : u \in \mathcal{A}(0) - \{v\}\}\mathbb{1}_{d_i=1} \cup \{v\}\mathbb{1}_{(d_i=1)\wedge([\min]-\mathcal{N}_v>1)} \cup \{v\}\mathbb{1}_{\mathcal{N}_v=[\max]}.$$

The final constraint is given by $\mathcal{C}_i = \bigcup_{v \in T} \mathcal{C}_i^{(v)}$. In this case, $\Omega = \{\mathcal{T}, [\min], [\max]\}$.

*Example.* Given $\mathcal{T} = \{x\}$, $[\min] = 1$, and $[\max] = 2$, and the partial sequence $[+, +, x, x]$ (corresponding to the expression $x + x + \square$), $x$ cannot be selected because it would exceed $[\max]$.

**Valency constraint.** This constraint can be enforced using the *length constraint*, since valency corresponds to arity. Thus, given the current value of $d_i$ and the valency value $n$, we can apply the *length constraint* with $[\min] = d_i + n$.

### A.2 Descriptions of existing priors

**Soft length prior.** The length constraint is known to result in highly skewed distribution over lengths (Kim et al., 2021b; Landajuela et al., 2021). To alleviate this phenomenon, Landajuela et al. (2021) propose combining the length constraint with a "soft" version that reduces the probability of terminal tokens early on (to discourage short sequences) and of

non-terminal tokens later on (to discourage long sequences). Specifically, for libraries with arities in $\{0, 1, 2\}$, they define the soft length prior at position $i$ as:

$$\ell_\circ = \left( \frac{-(i-\lambda)^2}{2\sigma^2} \mathbb{1}_{i<\lambda} \right)_{|\mathcal{A}(2)|} \| \, (0)_{|\mathcal{A}(1)|} \, \| \left( \frac{-(i-\lambda)^2}{2\sigma^2} \mathbb{1}_{i>\lambda} \right)_{|\mathcal{A}(0)|},$$

where $(\cdot)_n$ denotes that element $(\cdot)$ is repeated $n$ times, $\lambda$ and $\sigma$ are hyperparameters, and the tokens corresponding to logits $\ell$ are sorted by decreasing arity.

**Uniform arity prior.** With zero-initialized RNN weights, the initial distribution is uniform over all tokens in $\mathcal{L}$. One can convert this to instead to be uniform over all *arities* in $\mathcal{L}$ by applying a uniform arity prior (Landajuela et al., 2021) given by:

$$\ell_\circ = (-\log|\mathcal{A}(k)|)_{|\mathcal{A}(k)|} \| \cdots \| (-\log|\mathcal{A}(0)|)_{|\mathcal{A}(0)|},$$

where $k$ is the maximum arity of tokens in $\mathcal{L}$ and the tokens corresponding to logits $\ell$ are sorted by decreasing arity. This is typically combined with the soft length prior.

**Language model prior.** Kim et al. (2021a) introduce a *mathematical language model* trained on a "corpus" of mathematical expressions derived from Wikipedia, then use this language model as a prior to demonstrate an increase in search efficiency for symbolic regression. Given a language model $M$ that produces logits $\ell_M$ from a given partial sequence $\tau_{1:i-1}$, the language model prior is given by:

$$\ell_\circ = \lambda \ell_M,$$

where $\lambda$ is a hyperparameter controlling the strength of the language model prior, acting like an inverse temperature on the contribution of $\ell_M$ to the softmax computation.

