# OpenReview forum: "Incorporating domain knowledge into neural-guided search via in situ priors and constraints"
_ICML.cc/2021/Workshop/AutoML — AutoML@ICML2021 Poster_

### Official Review · Reviewer_2SPj · 2021-06-03
**Well written but lacking AutoML experiments**

**Rating:** 5
**Confidence:** 2

**Review:**

**Summary

Many methods (such as NAS) use autoregressive models when optimizing blackbox functions. This paper presents an approach to alter the output of such a model at an intermediate stage, in order to inject domain knowledge into the final output. Results are presented for a simple symbolic regression task.

My review score is borderline reject for two reasons: 1) The experiments are unrelated to AutoML 2) The motivation is how to inject domain knowledge, which goes against many AutoML works which seek to automate rather than add human knowledge. If there was an experiment using ML in a real setting it could be a good fit for this workshop. Without that, given my relatively limited knowledge of this area, I struggle to see the utility.

**Strengths
- The work is sufficiently broad that it could 1) aid others and 2) inspire future work.
- The paper is well written.
- If indeed there are settings where it could help with NAS, then that is an active area of research in AutoML.

**Weaknesses
- I am not convinced how useful the problem setting is. Do we really want a way to inject domain knowledge? Doesn't that go against the point of AutoML?
- The experiments are a toy symbolic regression, where the method presented trivially wins. Adding priors into a problem *should* make it perform better, but the challenge with AutoML is we often do not know this?

---

### Official Review · Reviewer_LnoL · 2021-06-13
**Preliminary work on an important topic**

**Rating:** 6
**Confidence:** 4

**Review:**

The paper addresses the problem of reducing the search space during NAS-type of searches, by introducing a set of rules and constraints that can prune the search. The method is evaluated on symbolic regression with promising results.

Incorporating rules and prior knowledge into search and learning methods is very important area of the study. The presented method does this through several proposed roles. The presentation is clear and the paper is well positioned within the field.

Overall, the idea is promising and is an appropriate workshop paper. To make the paper stronger, the authors should:
- Discuss the design choice of the selected priors. Why those? What are the limitations of the given set? What use cases they do not cover? And should we think about the selection of the priors in the pricipled way?
- Evaluate the method on more than one problem. And add some other baselines -- pruning techniques have been present for a while, and adding some heuristic and learned baselines would go along way to evaluate the approach' strengths and weaknesses.

---

### Decision · Program_Chairs · 2021-06-21

Accept (Poster)